# Bio-inspired Printed Monopole Antenna Applied to Partial Discharge Detection

**DOI:** 10.3390/s19030628

**Published:** 2019-02-01

**Authors:** Josiel do Nascimento Cruz, Alexandre Jean René Serres, Adriano Costa de Oliveira, George Victor Rocha Xavier, Camila Caroline Rodrigues de Albuquerque, Edson Guedes da Costa, Raimundo Carlos Silverio Freire

**Affiliations:** 1Post-Graduate Program of Electrical Engineering (PPgEE), Federal University of Campina Grande (UFCG), Aprigio Veloso 882, Campina Grande 58429-900, Brazil; adriano.oliveira@ee.ufcg.edu.br (A.C.d.O.); george.xavier@ee.ufcg.edu.br (G.V.R.X.); camila.albuquerque@ee.ufcg.edu.br (C.C.R.d.A.); 2Department of Electrical Engineering (DEE), Federal University of Campina Grande (UFCG), Aprigio Veloso 882, Campina Grande 58429-900, Brazil; alexandreserres@dee.ufcg.edu.br (A.J.R.S.); edson@dee.ufcg.edu.br (E.G.d.C.); rcsfreire@dee.ufcg.edu.br (R.C.S.F.)

**Keywords:** bio-inspired design, monitoring, partial discharge, printed monopole antenna, UHF, UWB

## Abstract

A new, bio-inspired printed monopole antenna (PMA) model is applied to monitor partial discharge (PD) activity in high voltage insulating systems. An optimized sensor was obtained by designing a PMA in accordance with the characteristics of the electromagnetic signal produced by PD. An ultra-wideband (UWB) antenna was obtained by applying the truncated ground plane technique. The patch geometry was bio-inspired by that of the *Inga Marginata* leaf, resulting in a significant reduction in size. To verify the operating frequency and gain of the PMA, measurements were carried out in an anechoic chamber. The results show that the antenna operating bandwidth covers most of the frequency range of PD occurrence. Moreover, the antenna presented a good sensitivity (mean gain of 3.63 dBi). The antenna performance was evaluated through comparative results with the standard IEC 60270 method. For this purpose, simultaneous tests were carried out in a PD generator arrangement, composed by an oil cell with point-to-plane electrode configurations. The developed PMA can be classified as an optimized sensor for PD detection and suitable for substation application, since it is able to measure PD radiated signals with half the voltage levels obtained from the IEC method and is immune to corona discharges.

## 1. Introduction

The insulation of high voltage equipment is subjected to stressful conditions, such as intense electric fields, chemical reactions, mechanical stresses, temperature variations and several environmental phenomena. In some situations, partial discharges (PDs) may occur, which are defined as low magnitude electrical discharges that partially short-circuit the insulation material [1]. The continuous occurrence of PDs can provoke a significant degradation of the material of the insulation, which can eventually lead to a full dielectric breakdown and, consequently, equipment failure. Therefore, the monitoring of PDs is a crucial tool to prevent the development of dielectric problems.

The most traditional method for PD detection is the one established by IEC 60270 [1], which allows the measurement of PD current pulses. This method, however, is considered highly invasive, since it requires the connection of a coupling capacitor in parallel with the monitored equipment, limiting its use to mainly laboratory applications.

In order to overcome the limitations imposed by the IEC standard method, researchers have been studying new techniques for PD detection. Among these techniques, the radiometric ones stand out for their non-invasive nature, since there is no need for an electrical connection between the sensor and the monitored equipment. Additionally, considering that the electromagnetic waves produced by PD present themselves in the UHF range (300 MHz to 3 GHz), UHF methods are immune to most of the interference that occurs in substations, such as those produced by corona discharges and power electronics switching, which are known to reach up to 300 MHz [2].

To improve the detection sensitivity of UHF sensors, researchers have studied the spectral characteristics of the signals radiated by PD, concluding that, generally, most of the radiated energy is concentrated between 300 MHz and 1.5 GHz [2,3,4]. The first application of UHF sensors to PD detection is reported in [5], attesting the efficiency of the UHF method for the detection and location of PD in a Gas Insulated Substation (GIS). Over the years, others have proved the efficiency and reliability of the UHF method in a GIS [6,7,8], and its application was extended to other high voltage equipment, i.e., mainly power transformers and high voltage cable connections [9,10,11].

Among the different types of UHF sensors, printed monopole antennas (PMA) are notable for having desirable characteristics for practical applications, such as attractive radiation patterns, low cost, wideband and their ease of installation and manufacture [12]. Printed monopole antennas with traditional radiating element (patch) shapes and designed to operate in UHF range would assume relatively large dimensions, limiting their use for practical application in PD detection.

The operating frequency of PMA is directly related to the perimeter of the radiating element. The greater the perimeter, the lower the operating frequency. Therefore, techniques used for antenna miniaturization mainly seek optimized geometries that maximize the patch perimeter/area ratio.

These optimized geometries can be found in both fauna and flora. The different shapes present in organisms of nature are the result of thousands of years of evolution, providing great efficiency regarding the ability to survive. Hence, aiming to achieve a better radiating efficiency, bio-inspired antennas use the structure of plants or animals as a basis for their design. Moreover, bio-inspired designs generally present a greater perimeter/area ratio, allowing a reduction of size of the antenna. 

Studies involving bio-inspired antennas have been receiving great emphasis, among these, [13,14,15,16,17,18] can be highlighted. In [13], the authors developed a printed monopole antenna with maple-leaf geometry, operating in ultra-wideband (UWB). In [14], the authors designed a PMA bio-inspired by a butterfly, for operation in UWB, with circular slots in the patch for band-stop purposes. A coplanar waveguide antenna developed in [15] was bio-inspired by bee antennas, operating in 2.4 GHz. An antenna for UWB applications using cotton leaf inspired geometry was developed in [16], with a bandwidth of 3–10 GHz. A gyngko biloba bio-inspired PMA was developed in [17], with fabric substrate (denim), operating in the 2G, 3G and 4G bands, with measurements taken close to the body and a gain of 3 dBi. In [18], a gielis superformula was applied to design several leaf-shaped PMA optimized for wireless applications.

Although some wideband planar antennas and fractal structures provide smaller shapes [19,20,21,22,23,24], none of them achieves the fully coverage of the PD activity frequency range. Most of them address multi-narrow-band operation [19,20,21], reach only modestly performance (Voltage Standing-Wave Ratio [VSWR] < 5) [22,23,24] or only work at higher frequency bands (0.75–1.5 GHz) [25]. Then, with the simultaneous application of a bio-inspired geometry design and bandwidth enhancement techniques, it is expected the development of a new UHF sensor that presents a compact size and enough bandwidth for the PD detection application, providing a better relation size-bandwidth than the cited and other antennas found in the PD application literature [26,27,28,29,30,31,32]. Additionally, bio-inspired geometries provide higher density of current concentrations on the antenna transmission line, resulting in higher mean gain than classical and modified PMA shapes developed for PD detection [20,21,26,27,31], resulting in a higher PD detection sensitivity. 

Finally, in order to apply the UHF method in open area environments, mainly substations, it is essential that the applied antennas have an omnidirectional pattern. However, the radiation patterns for wideband antennas applied in PD detection usually suffer distortions as the frequency increases within the PD activity range, as presented in [20,26,28,29,31,32]. Hence, the development of a bio-inspired antenna that provides an omnidirectional and constant radiation pattern for entire range of PD activity frequency is also sought in this work. 

Therefore, in the present paper, a new bio-inspired PMA model that addresses optimal bandwidth, size, gain, radiation pattern and sensitivity for PD detection is proposed. For this purpose, the truncated ground plane technique was used to enhance the antenna’s bandwidth, achieving a UWB antenna that covers most of the band of interest (300–1500 MHz). Furthermore, a bio-inspired geometry based on the *Inga Marginata* leaf was applied to the patch, resulting in a significant reduction in size and gain increase. Finally, the designed antenna was subjected to bandwidth and gain measurement tests (in an anechoic chamber) and sensitivity tests of PD detection by means of comparative results with the standard IEC 60270 method, applied in a PD generator arrangement, composed of an oil cell with point-to-plane electrode configuration, attesting the developed PMA bio-inspired sensor applicability in PD detection.

## 2. Printed Monopole Antennas

PMA have a simple structure and are easily manufactured. Moreover, features such as an omnidirectional radiation pattern and large bandwidth make this type of antenna suitable for UWB applications. A basic PMA structure is shown in Figure 1, where *W_0_* and *L_0_* are the antenna length and width, respectively, *L* and *W* are the length and width of the patch, respectively, *L_g_* and *W_g_* are the length and width of the truncated ground plane, respectively, *W_f_* is the width of the transmission line, *g* is the distance between the patch and the ground plane, and *h* is the thickness of the substrate. To achieve a better impedance matching, a slot can be inserted on the top of the ground plane [33], its length and width being denoted by *L_s_* and *W_s_*, respectively.

The lower frequency of the operating band of a PMA can be approximated depending on the patch perimeter, as presented in Equation (1), where *p* is the perimeter of the antenna patch and ε_ref_ is the effective relative permittivity of the dielectric [34]. The electric current density in a PMA is greater at the edges of the patch. Therefore, an increase of the patch perimeter would also increase the wavelength and, consequently, decrease the lower operating frequency [13].
*f* (GHz) = 300/(p.√ε_ref_),(1)
ε_ref_ = (ε_r_ + 1)/2 + (ε_r_ − 1)/2.(1 + 12.*h*/*W*)^−1/2^, (2)
where *W/h* > 1, and *h* represents the dielectric substrate thickness and *W* the microstrip width. 

## 3. Material and Methods

The High Frequency Structure Simulator (HFSS) from the ANSYS Electronics Desktop software package was used for the simulation and design of the antenna. In all the simulations, the substrate is a low-cost fiberglass (FR-4) with dielectric constant (ε_r_) equal to 4.4, thickness (*h*) of 1.6 mm and loss tangent (δ) equal to 0.02.

The simulated antenna bandwidth was defined as the entire frequency range at which the reflection coefficient values (S_11_) were below the −10 dB threshold. In addition, the gain values were also extracted from the simulated design, since it has been reported in the literature that antennas used for PD detection should have mean gain higher than 2 dBi for the operating bandwidth of interest [35]. Lastly, the radiation patterns of the designed bio-inspired antenna were extracted.

After the simulation stages, the designed antenna was built and subjected to reflection coefficient and gain measurements in an anechoic chamber. Then, the antenna was applied in a PD generator experimental arrangement in order to estimate the antenna sensitivity regarding the conventional method for PD detection, the IEC 60270 standard.

In the following subsections, the designed bio-inspired antenna and the experimental procedures for the tests to measure the reflection coefficient, gain, and sensitivity are detailed. 

### 3.1. Bio-Inspired Antenna Design

The bio-inspired design applied in this work is based on the arrangement of the leaves of the tropical plant scientifically named *Inga Marginata Willd* [36], presented in Figure 2.

From the PMA model shown in Figure 1 and the perimeter Equations (1) and (2), the patch was designed to present the geometry of the proposed leaf and UHF range (300–3000 MHz). For this, the length of the feed line was defined as λ/4 of the first resonance frequency (300 MHz), i.e., 208 mm. For the 50 Ω impedance matching, the microstrip line was designed with a width of 3 mm. In order to obtain a higher bandwidth and reflection coefficient, a slit with dimensions 3.2 × 35 mm was inserted at the central top of the ground plane. In order to obtain an optimized structure, geometric parameter sweeps, such as width and length of the leaves, as well the distance and opening angle between them, were performed. The detailed dimensions of the described bio-inspired antenna are summarized in the schematic presented in Figure 3.

After the design stage in the simulation environment, the bio-inspired antenna was built using the same simulated specifications and is presented in Figure 4.

### 3.2. Laboratory Tests

The manufactured bio-inspired antenna presented in Figure 4 was subjected to gain, reflection coefficient and sensitivity tests as described in the following.

#### 3.2.1. Reflection Coefficient and Gain Measurements

The reflection coefficients were obtained through an ENA E5062A network analyzer of Agilent Technologies. For the gain measurement, an experimental arrangement composed by a reference antenna with known gain and the bio-inspired antenna was used as presented in Figure 5.

The distance *R* between the antennas, shown in Figure 6, represents the far field distance for the constructed antenna. The selected far field distance was defined regarding the maximum frequency on which the PD pulses still present significant irradiated energy, i.e., 1500 MHz [2,3,4]. Therefore, from the dimensions of this bio-inspired antenna and the far field equations [12], the calculated value of *R* was found to be 1.5 meters. 

As reference antenna, a Hyperlog 30100X log-periodic antenna, from Aaronia AG, with operating frequency range 380 MHz–10 GHz and 4.5 dBi gain was used.

From the measured values of the transmitted (reference antenna) and received (bio-inspired antenna) powers (*P_T_* and *P_R_*, respectively) and reference antenna gain (*G_R_*), the gain of the bio-inspired antenna (*G_D_*) was obtained by using the Friis equation [38]:
G_D_ (dBi) = P_R_ (dBm) − P_T_ (dBm) − G_R_ (dBi) + 20log(4πR/λ) (dBi).(3)

In order to reduce the effect of external interference and signal reflections during the tests, the reflection coefficient and gain measurements were carried out in an anechoic chamber. Figure 6 presents the practical gain arrangement used in the anechoic chamber.

#### 3.2.2. PD detection sensitivity test

For the PD detection test, the bio-inspired antenna was positioned at the selected far field distance (1.5 meters) from the applied PD generator, composed by an oil cell with point-to-plane electrode configuration spaced by 2.0 cm. To estimate the antenna PD detection sensitivity using conventional techniques, the IEC 60270 method was applied simultaneously to the circuit, employing a coupling capacitor (1000 pF) and a measuring impedance (LDM-5, from Doble Lemke). The PD pulses were acquired in the time and frequency domains. For the time domain, the measurements were carried out using a Keysight oscilloscope DSO90604A with bandwidth of 6 GHz, sampling of 20 GSa/s, rise time of 70 ps and 4 analog channels. For the frequency domain measurements, a vector network analyzer (VNA) E5071C (9 kHz–8.5 GHz) from Keysight Technologies was used. A schematic and a real size photo of the described arrangement are presented in Figure 7 and Figure 8, respectively.

In order to obtain the antenna PD detection sensitivity in terms of the apparent charge, all the calibration procedures established by the IEC 60270 standard were executed before applying the voltage in the PD generator oil cell. For this, the LDC-5 calibrator from Doble Lemke was applied at the oil cell terminals, and the pulses were collected by measuring the impedance. The calibration data for three values of the apparent charge (20, 100 and 500 pC) were collected and are presented in Table 1.

From the calibration data presented in Table 1, it is possible to verify an approximately linear relation between the apparent charge and the measured voltage values. Therefore, the PD values of the apparent charge of the PDs generated in the oil cell can be estimated from the voltages measured by the IEC 60270 method during the application of the high voltage. Then, the antenna PD detection sensitivity in terms of the apparent charge may be evaluated.

## 4. Results

The analysis of the results is divided into two subsections. The first is about the antenna characterization results (simulated/measured reflection coefficient and gain), while the second subsection is about the antenna PD detection sensitivity.

### 4.1. Reflection Coefficient and Gain

Figure 9 shows a comparison between the measured and simulated reflection coefficients for the bio-inspired PMA.

Both measured and simulated results presented similar behavior and a large bandwidth, starting around 340 MHz up to frequencies above 8 GHz. Therefore, the bio-inspired antenna is capable of PD detection in the applications, since it covers the full frequency range of PD activity (300–3000 MHz).

Figure 10, Figure 11 and Figure 12 show the 2D and 3D simulated radiation patterns, respectively, of the bio-inspired antenna for the frequencies of 350 MHz, 900 MHz and 1500 MHz. For the observed frequencies, a maximum directive gain of 2.33 dBi, 4.08 dBi and 4.68 dBi, respectively, was obtained. The frequencies selected for analysis are within the range of interest for the monitoring of partial discharges (300–1500 MHz), since more significant energy concentrations are reported in this frequency range.

From the obtained radiation patterns, it can be verified that besides the UWB characteristic, the use of truncated ground plane and Inga Marginata bio-inspired geometry resulted in an antenna with omnidirectional behavior throughout the operating bandwidth. This kind of radiation pattern allows its application to PD detection in open area environments, such as substations, detecting PD signals originating from sources in any direction and for several pieces of equipment. In addition, omnidirectional behavior allows the use of PD source location algorithms.

Usually, the gain of a PMA is around 2–3 dBi. However, the proposed Inga Marginata bio-inspired geometry provides, besides the dimensional reduction, a higher density of current concentration on the patch boundaries, as shown in Figure 13, resulting in a higher gain than general PMA structures and other bio-inspired models presented in the literature.

Using the procedures exposited in the previous section in an anechoic chamber, the bio-inspired PMA gain was measured. The comparison between the measured and simulated gain is presented in Figure 14.

The mean gain calculated for the measured and simulated curves are 3.63 dBi and 3.58 dBi, respectively. These values are higher than the 2 dBi average limit for PD detection recommended by [35]. Thus, the bio-inspired PMA has enough sensitivity for PD detection. 

From the simulated/measured reflection coefficient and gain results presented in this subsection, the bio-inspired PMA can be considered fit for the detection of PD. Hence, it was subsequently submitted to detection sensitivity tests, as presented in the following subsection.

### 4.2. PD Detection Sensitivity

The beginning of PD activity was detected simultaneously by the bio-inspired antenna and the IEC 60270 standard method for the application of 28.2 kV in the oil cell. Samples of the detected PD pulses are presented in Figure 15.

The antenna was able to detect all the PD pulses detected by the IEC 60270 method. However, the antenna’s sensitivity is significantly lower than the IEC 60270 method, presenting millivolt values reduced by one-half compared to the standard method. This resulted is expected, since the radiated waves are susceptible to higher losses, such as reflections and propagating attenuations, than those that occur in the standard method, which consists of a direct electrical connection with the monitored equipment, presenting lower losses.

From the calibration data presented in Table 1, the apparent charge generated by the PD pulses presented values between 65 and 498 pC. This range of apparent charges represents a good approximation of the PD low intensity pulses detected in practice [39]. Then, despite the lower magnitudes of the values compared to the standard method, the antenna’s detection sensitivity is validated, since it was able to detect PD pulses with apparent charges corresponding to the inception of an insulating problem (low intensity pulses).

The PD pulses presented in Figure 15 were also analyzed in the frequency domain, as presented in Figure 16. 

In order to highlight the PD frequencies of occurrence, the background noise of the PD measurement setup was recorded during 30 minutes before the high voltage application. Due to the absence of a shielded high voltage laboratory, some digital television UHF interference (532.5, 693.5, 775.5 and 882.5 MHz) was detected and considered as background noise. Then, any significant changes in the background noise curve was considered as partial discharges. Therefore, the frequency domain results show that most of the generated PD pulses detected by the bio-inspired PMA were concentrated in the frequency range of 1–1.1 GHz and around 725 MHz. These frequency ranges are in agreement with the values reported in [2,3,4] as the principal range of interest for the monitoring of partial discharges (300–1500 MHz). In addition, the large bandwidth presented by the PMA eases the detection of different frequency patterns generated by several types of PD in different frequency ranges, such as internal and superficial defects. 

#### Special Case: Corona Detection

Corona discharges are a common phenomenon in high voltage equipment, transmission lines, and substations. The occurrence of corona produces electromagnetic waves with significant energy components with frequencies up to 200–300 MHz [2]. Therefore, corona discharges are one of the main sources of interference for radiometric based methods. Hence, for an investigation of the immunity to corona interference of the bio-inspired PMA, the experimental arrangement was subjected to high voltages until corona discharges were generated on the circuit connections and were detected by the IEC 60270 standard method.

According to [40,41], the difference between PD and corona discharges can be defined by the location pulse phase compared with the sinusoidal reference voltage curve. The occurrence of a corona can be proven by the presence of pulses located at the positive or negative peak values of the sinusoidal reference voltage curve, whereas PD discharges are located in proximity to the semicycle transitions. Based on this, the detected pulses were distinguished and the detection of the inception of a corona discharge was achieved for a 34.3 kV application, as presented in Figure 17.

From Figure 17, it can be seen that the antenna retained its detection sensitivity regarding the PD pulses located near the semicycle transitions. However, only the IEC 60270 standard method was able to detect a corona occurrence at the positive peak of the sinusoidal reference voltage. Therefore, the bio-inspired PMA presented itself as insensitive to corona occurrence. Hence, its immunity to corona interference reinforces the applicability of PMA to PD detection in substations, since the antenna is immune to its main interference source. In addition, this immunity provides better signal to noise ratios and, consequently, needs less effort in the application of filtering and data processing techniques.

## 5. Conclusions

In this paper, a bio-inspired PMA model based on the *Inga Marginata* leaf for PD detection was proposed. The designed antenna has the potential to be applied as a UHF sensor for PD detection, since it meets the bandwidth (300–1500 MHz) and gain (>2 dB) requirements according to the laboratory experiments carried out in an anechoic chamber. In high voltage tests, the potential of the antenna for PD detection was attested; it was shown to be able to detect several PD signals generated in a point-to-plane electrode configuration. By evaluating the induced voltage in the PMA terminals, the PD intensity and evolution could be assessed. Therefore, the proposed PMA can provide continuous, non-invasive, and relatively low-cost monitoring of PD activity in high voltage insulating systems. The conclusions obtained from the results of this article can be summarized as follows:(1)The application of the *Inga Marginata* leaf geometry to a PMA resulted in a UWB sensor with operating bandwidth (340 MHz–8 GHz) that covers almost the entire frequency range of PD activity (300–1500 MHz);(2)The implementation of the *Inga Marginata* geometry resulted in an omnidirectional behavior, allowing the application to PD detection in open area environments (such as substations) and favoring the employment of PD source location algorithms;(3)Through measurements in an anechoic chamber and the Friis formulation, the calculated mean gain for the bio-inspired PMA was equal to 3.63 dBi, representing a good detection sensitivity for PD application;(4)In PD laboratory tests, the bio-inspired PMA was able to detect PD pulses with apparent charges above 65 pC, generated in a point-to-plane electrode configuration immersed in an oil cell;(5)The bio-inspired PMA presented a measured voltage with half the magnitude obtained from the IEC 60270 standard method, resulting in a PD detection sensitivity that is good for a radiometric based method;(6)Lastly, the bio-inspired antenna presented itself as immune to corona discharges, which is the main source of interference for monitoring techniques in substations, which is an additional feature that emphasizes the antenna’s potential for PD detection.

From the obtained results, it can be concluded that the developed PMA can be classified as an optimized sensor for PD detection and suitable for substation applications, since it meets the frequency, gain, detection sensitivity and immunity requirements associate to PD measurement applications.

## Figures and Tables

**Figure 1 sensors-19-00628-f001:**
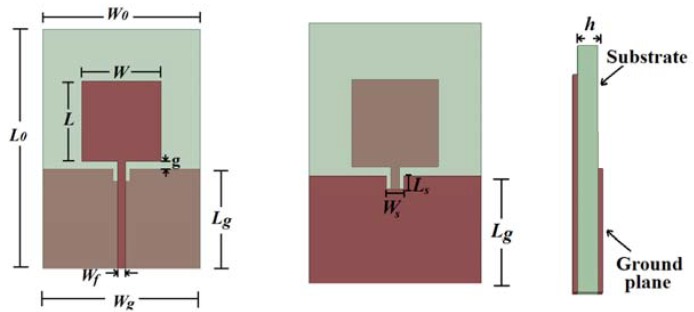
Geometry of a rectangular printed monopole antenna.

**Figure 2 sensors-19-00628-f002:**
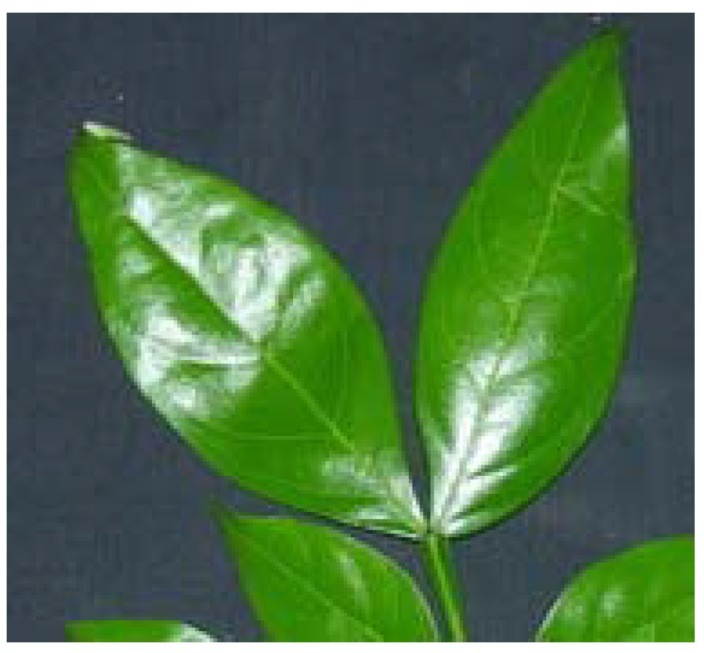
*Inga Marginata* leaves [37].

**Figure 3 sensors-19-00628-f003:**
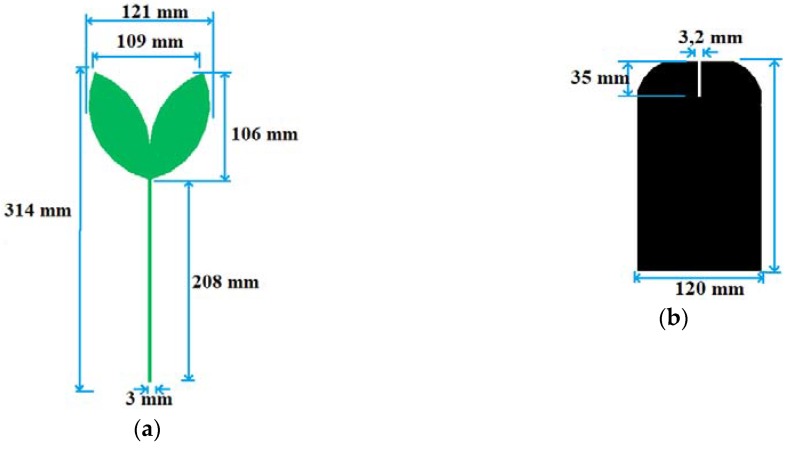
Designed model for the *Inga Marginata* bio-inspired PMA antenna: (**a**) Patch dimensions; (**b**) Ground plane dimensions.

**Figure 4 sensors-19-00628-f004:**
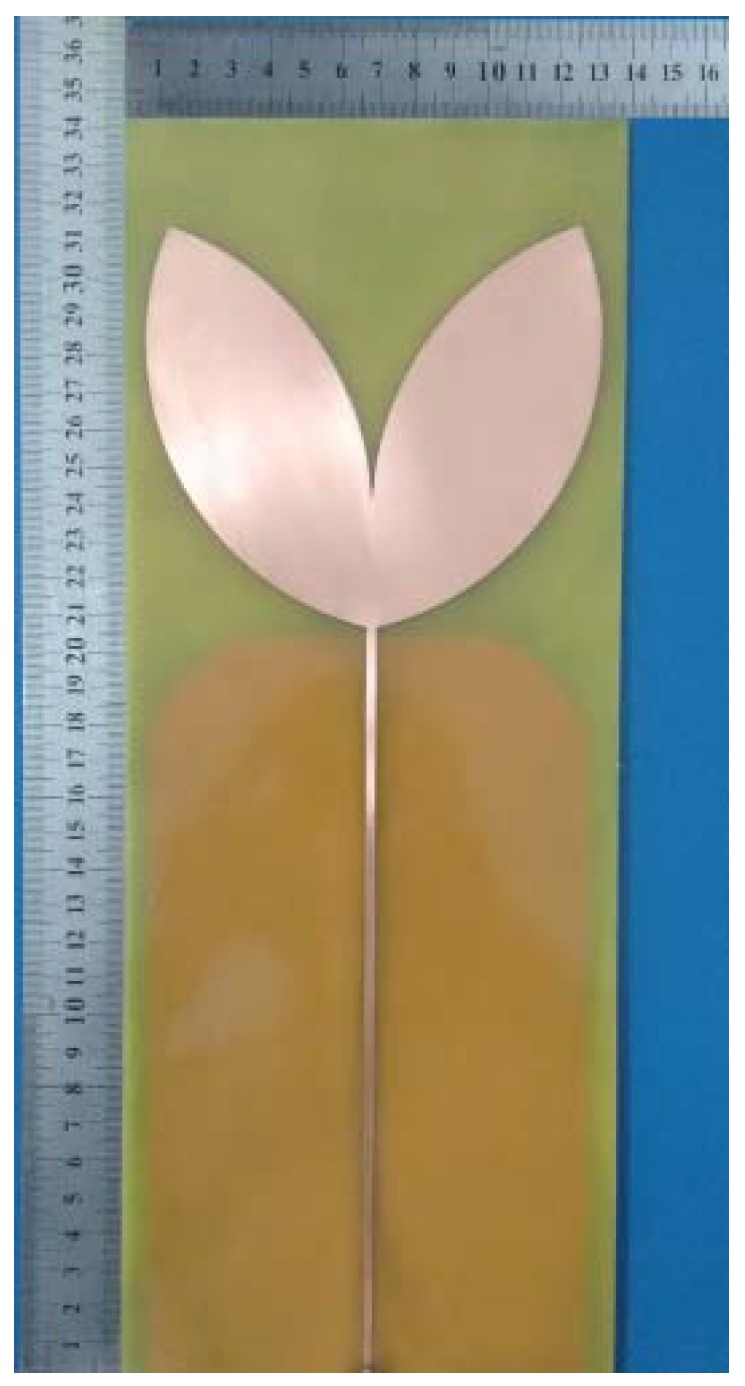
Manufactured bio-inspired antenna.

**Figure 5 sensors-19-00628-f005:**
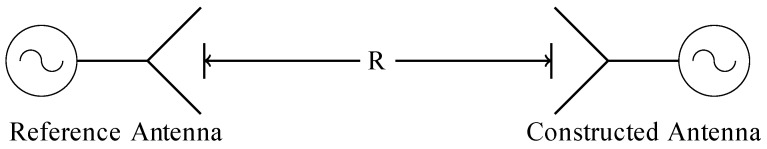
Experimental arrangement for the gain measurement.

**Figure 6 sensors-19-00628-f006:**
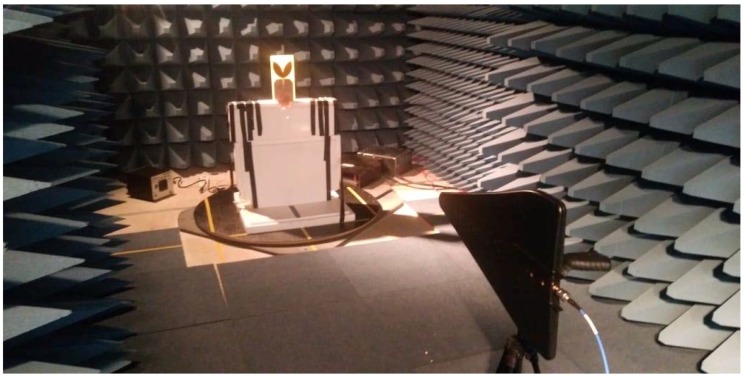
Gain measurement setup in the anechoic chamber.

**Figure 7 sensors-19-00628-f007:**
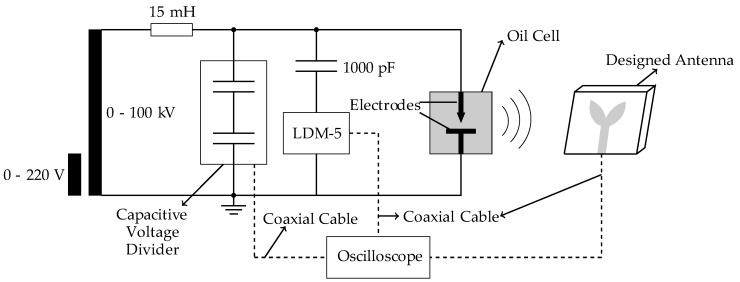
PD detection sensitivity test.

**Figure 8 sensors-19-00628-f008:**
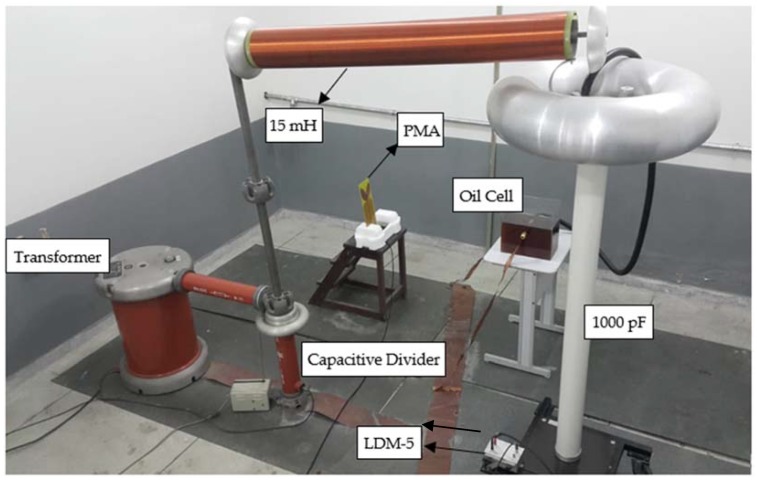
PD detection sensitivity test setup.

**Figure 9 sensors-19-00628-f009:**
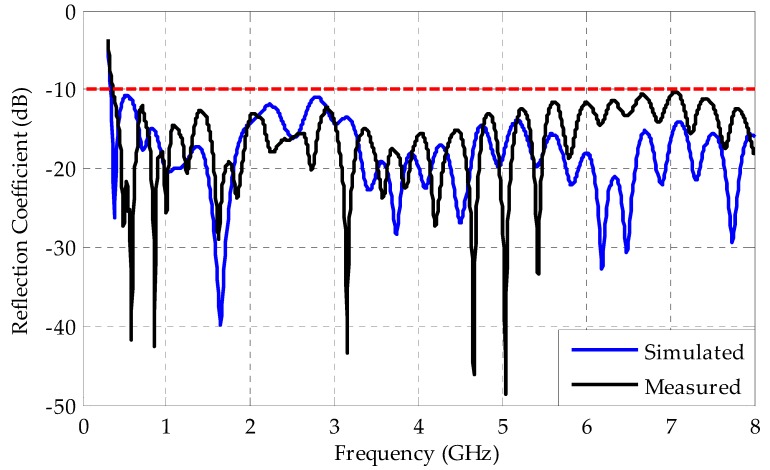
Measured and simulated reflection coefficients for the bio-inspired PMA.

**Figure 10 sensors-19-00628-f010:**
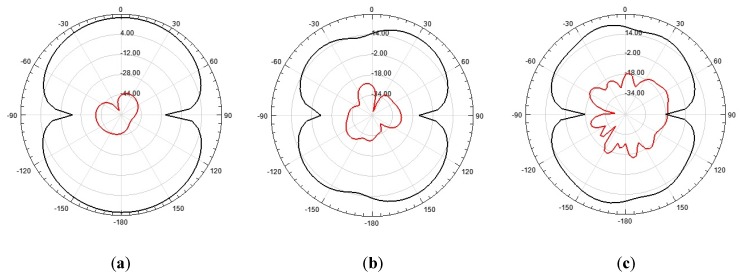
Bio-inspired PMA E-plane co-polarization (black) and cross-polarization (red) radiation patterns, respectively: (**a**) 350 MHz; (**b**) 900 MHz; (**c**) 1500 MHz.

**Figure 11 sensors-19-00628-f011:**
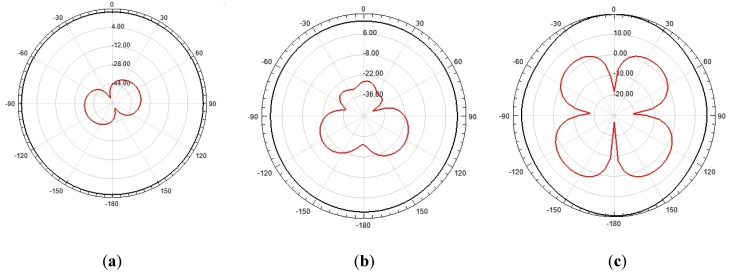
Bio-inspired PMA H-plane co-polarization (black) and cross-polarization (red) radiation patterns, respectively: (**a**) 350 MHz; (**b**) 900 MHz; (**c**) 1500 MHz.

**Figure 12 sensors-19-00628-f012:**
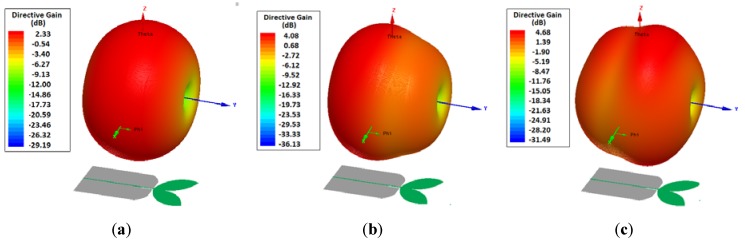
PMA bio-inspired 3D radiation patterns: (**a**) 350 MHz; (**b**) 900 MHz; (**c**) 1500 MHz.

**Figure 13 sensors-19-00628-f013:**
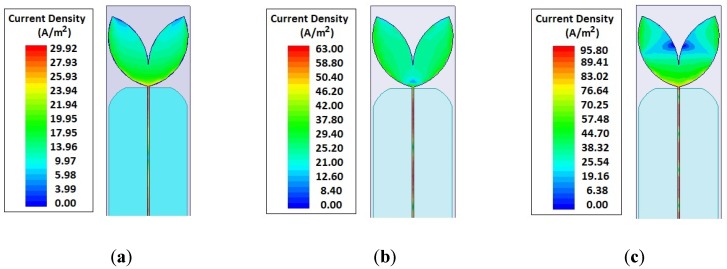
PMA bio-inspired electric current distributions: (**a**) 350 MHz; (**b**) 900 MHz; (**c**) 1500 MHz.

**Figure 14 sensors-19-00628-f014:**
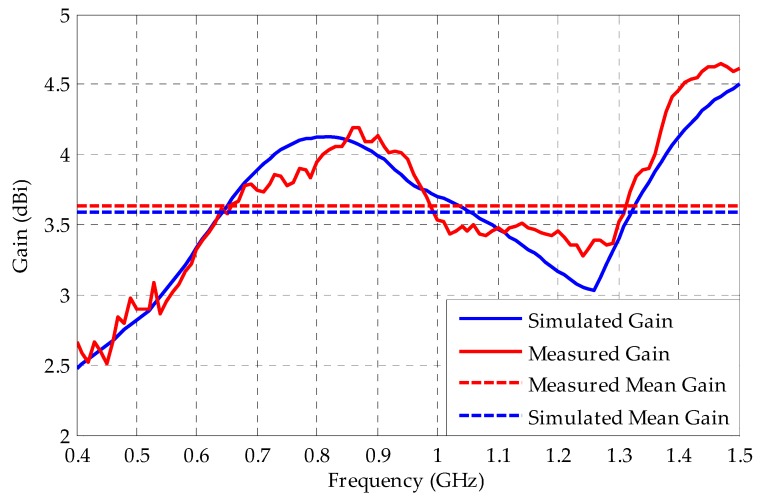
Comparison between the measured and simulated gain and their respective calculated mean gain for the bio-inspired PMA.

**Figure 15 sensors-19-00628-f015:**
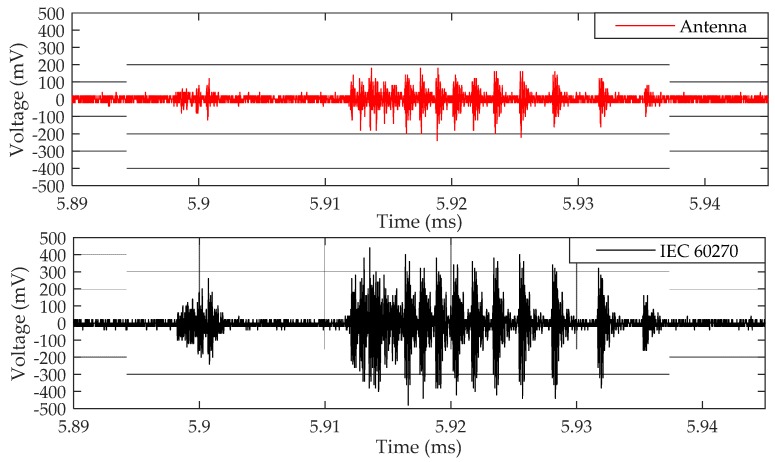
Sample PD pulses detected by the bio-inspired PMA and standard method for point-to-plane configuration at 28.2 kV.

**Figure 16 sensors-19-00628-f016:**
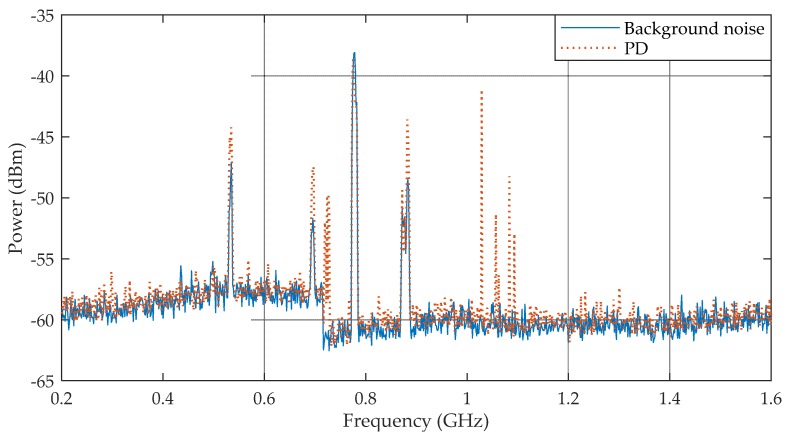
Samples of PD pulses in the frequency domain.

**Figure 17 sensors-19-00628-f017:**
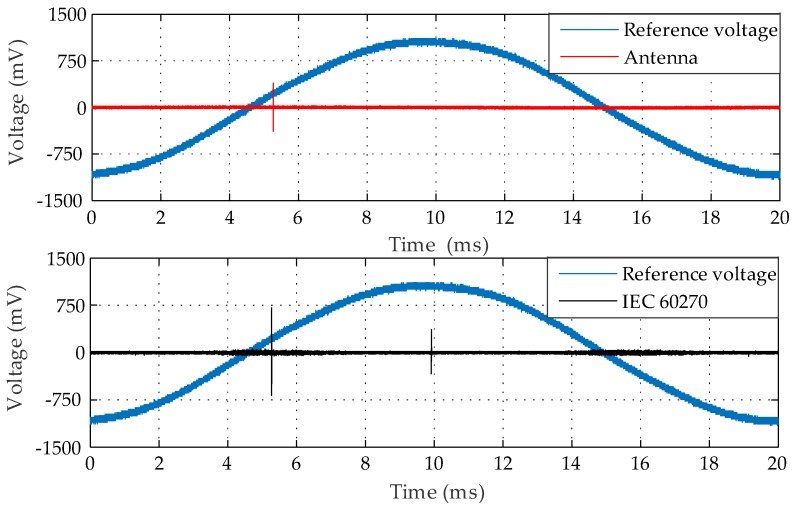
PD and corona pulses for point-to-plane configuration at 34.3 kV.

**Table 1 sensors-19-00628-t001:** Calibration apparent charge data for the applied PD generator oil cell.

Apparent Charge (pC)	Voltage (mV)
20	23.6
100	98
500	480

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
