# Peer review of "Bio-inspired Printed Monopole Antenna Applied to Partial Discharge Detection"

_sensors, 2019, doi:10.3390/s19030628_

Round 1
Reviewer 1 Report
This is a very interesting and well-written paper that is certainly worth of publication. I would like to point out some points that would improve the quality of the manuscript.
Antenna gain should be in dBi (relative to a hypothetical isotropic antenna).
I am not sure about the validity of eqs (1) and (2). Maybe further refs apart from [13]
should be included and it should be pointed out that they are approximately valid in this case (especially eq. 2).
In the antenna design section, it should be mentioned, in my opinion, if any simulation optimization attempts or geometric parameter sweeps have been undertaken.
From Fig. 6 it is seen that the horizontal polarized component of the antenna gain was measured. However, these antennas tend to be vertically polarized. Did you also measure
with the reference antenna turned by 90 degrees so that its dipoles are vertical ?
In Fig. 9 it seems that there are some connecting cable problems with the measured antenna (deep nulls and repeated pattern above 4GHz). Could you please check again the connecting cables.
In Fig. 10 it would be good to have both E-plane and H-plane antenna patterns (azimuth and elevation).
In Fig. 12, the simulated antenna gain is missing. An antenna gain around 10 dBi at 1.5 GHz
seems to be excessive. Is this boresight gain from measurements ? In this case the radiation pattern should be much more directive.
Author Response
Thank you for all your comments. Due to the need of the modification of some Figures in the paper, as suggested by you, the point-to-point responses to your comments are presented in the word document attached.

Reviewer 2 Report
The paper presents the use of a wideband planar monopole for detection of partial discharge. The reviewer's comments are as follows:
1-The first part of the paper is redundant and not novel. The first 5 pages can be removed and reduce to a short introduction. The part of introduction that relates to planar monopole antenna and microstrip antennas is sometimes misleading and not correct. For example, the paper and all the references are presenting planar monopole antennas. Planar monopole antennas are not the same as microstrip patch antennas. In the middle of introduction, the authors start to talk about the limitations of microstrip patch antennas, which is irrelevant here. In addition, the comment that microstrip antennas are known for their narrow bandwidth is not correct. There are many designs of microstrip patch antennas that have overcome these limitations. The presented design of monopole antenna has been previously published by the same authors, therefore, pages 1-5 do not present much novel information.
2-This statement:
“Optimized geometries can be found in both fauna and flora. The different shapes present in organisms are the result of thousands of years of evolution, providing great efficiency regarding the ability to survive. Hence, aiming to achieve a better radiating efficiency, bio-inspired antennas use the structure of plants or animals as a basis for their design.” is confusing and not exactly true. Some bio-inspired shapes present larger circumference in smaller size (such as fractal shapes), however, the shape presented in this paper is not really taking much advantage of that, except that it looks like a leaf. The authors did not show why a leaf shape is a better shape than any other shapes that have been presented in many published papers for ultra-wideband monopole antennas.
3-Equation (1) is not clear, what is “p.”? Equation (2) is wrong, since it does not consider substrate thickness. It can only be used for approximation.
4-The measurement method in the anechoic chamber is a well-known procedure and there is no need to explain it, this takes about one page of the paper.
5-Fig. 9 shows the reflection coefficient, not return loss. Return loss is positive.
6-Fig. 10, co-pol and cross-polarization patterns should be shown. Planar monopoles usually suffer from high cross-polarization. With the proposed design, the cross-polarization level might be very high.
7-Fig. 12, should be compared with the simulated gain. The gain of monopole is usually around 2-3 dBi, while the measured gain is over 6dBi. Please explain.
8-Fig. 13, please explain the setting of the oscilloscope used for measuring the antenna voltage. Was there any attempt to move the antenna to other locations to see the effects of multipath reflections? Why this antenna is being considered for this measurement? Was there any comparison for example with a standard gain horn antenna?
9-Please explain the detail of the analysis done to generated Fig. 14.
10-In Fig. 15 the level of the first detection of the antenna and IEC 60270 were almost the same, while in Fig. 13 it was almost half. Why is it different this time? Again the setting of measuring oscilloscope is not given. Was this experiment done only once? If yes, in your opinion, having one measurement that could not detect corona pulse is enough to conclude the antenna is not able to do this detection?
Overall, although the concept is interesting, the paper needs to be revised to remove non-novel parts and expand the details of the PD measurements. Comparison with another reference antenna can be useful to show why the proposed design is superior.
Author Response

(The authors gave the same response as above.)

Round 2
Reviewer 2 Report
Thanks for responding to the comments. There are still a few of comments:
1-It seems that the cross polarization is only shown for E-plane. It is better instead of repeating E-plane co-pol and having E- and H-plane patterns together to show E-plane co and cross-pol in one figure and H-plane co and cross-pol in another figure.
2-The results of gain is shown for up to 1.5 GHz. The antenna measurement and simulations were done up to 8 GHz. Why gain is not shown for higher frequency range?
3-It is still not clear why this antenna is used for PD detection. There are many options of planar wide band antennas that are low-cost and small. Why this one?
4-It is preferred if the performance of this antenna is compared with a well-known wide band antenna as a reference. The results of PD detection are only given for the proposed antenna. It can not be compared with any reference. Conclusions can be drawn more confidently if a comparison was done.
Author Response

(The authors gave the same response as above.)
